# Building an annotated corpus for automatic metadata extraction from multilingual journal article references

**Wonjun Choi, Hwa-Mook Yoon, Mi-Hwan Hyun, Hye-Jin Lee, Jae-Wook Seol, Kangsan Lee, Dajeong Lee, Young Joon Yoon, Hyesoo Kong** *

Digital Curation Center, Korea Institute of Science and Technology Information, Daejeon, Republic of Korea

* hyesoo@kisti.re.kr

**Data Availability Statement:** All data are available from http://doi.org/10.23057/47.

**Funding:** This research was supported by the Korea Institute of Science and Technology

## Abstract

Bibliographic references containing citation information of academic literature play an important role as a medium connecting earlier and recent studies. As references contain machine-readable metadata such as author name, title, or publication year, they have been widely used in the field of citation information services including search services for scholarly information and research trend analysis. Many institutions around the world manually extract and continuously accumulate reference metadata to provide various scholarly services. However, manually collection of reference metadata every year continues to be a burden because of the associated cost and time consumption. With the accumulation of a large volume of academic literature, several tools, including GROBID and CERMINE, that automatically extract reference metadata have been released. However, these tools have some limitations. For example, they are only applicable to references written in English, the types of extractable metadata are limited for each tool, and the performance of the tools is insufficient to replace the manual extraction of reference metadata. Therefore, in this study, we focused on constructing a high-quality corpus to automatically extract metadata from multilingual journal article references. Using our constructed corpus, we trained and evaluated a BERT-based transfer-learning model. Furthermore, we compared the performance of the BERT-based model with that of the existing model, GROBID. Currently, our corpus contains 3,815,987 multilingual references, mainly in English and Korean, with labels for 13 different metadata types. According to our experiment, the BERT-based model trained using our corpus showed excellent performance in extracting metadata not only from journal references written in English but also in other languages, particularly Korean. This corpus is available at http://doi.org/10.23057/47.

## Introduction

Bibliographic references are citations of previous studies that authors refer to while conducting their own studies. These references typically appear at the end of scientific articles. They contain valuable meta-information such as the author name, title, journal name, and publication

Information (KISTI) of the Ministry of Science and ICT, South Korea (MSIT) (No. K-23-L01-C01: Construction on Intelligent SciTech Information Curation). The funders had no role in study design, data collection and analysis, decision to publish, or preparation of the manuscript. There was no additional external funding received for this study.

**Competing interests:** The authors have declared that no competing interests exist.

year, also known as "*metadata.*" As references serve as a vital link between previous and latest studies, collecting such metadata from references is an essential step in the development of autonomous citation-indexing systems such as Google Scholar [1], SCOPUS [2], Web of Science [3] and PubMed [4]. These systems help researchers effectively search scientific information via intelligent information retrieval and recommendation systems, which require a large amount of machine-readable metadata collected from scientific articles. Because the number of articles published annually has grown exponentially in recent decades [5–7], there is a significant demand for automated methods and tools that enable researchers to automatically extract high-quality bibliographic metadata directly from the raw reference.

Bibliographic reference parsing is the process of extracting bibliographic components from individual references. It is useful for identifying cited articles, a process known as citation matching [8]. Citation matching is the most critical requirement in determining the impact of journals [9, 10], researchers [11] and research institutions [12], and in assessing document-to-document similarity [13]. In reference parsing, a raw reference string following a specific style is used as input. The output is a machine-readable parsed reference that is composed of a metadata field type (*e.g.*, "page") and a value (*e.g.*, "394–424"), as described in Fig 1.

Various open-source reference parser tools are currently available. In their earlier versions of the reference parser tools such as BibPro [14], Citation [15], and Citation-Parser [16], regular expressions, handcrafted rules, and template matching were used. Regular expressions are a traditional method for approaching the reference parsing task. It involves capturing the patterns of metadata field values in the reference texts based on the defined expressions for different reference styles. Typically, reference parser tools using regular expressions are known to be effective when the given reference matches one of the defined expressions. In template matching, the references are first matched against the pre-defined templates, representing most citation formats, and then template-specific rules or regular expressions are used to extract the metadata field values. However, as these methods depend on pre-defined rules, templates, and regular expressions, they may perform poorly when undefined references are provided.

Unlike the above-mentioned approaches, in a supervised machine learning (ML)-based approach, a model learns how to classify metadata directly from the training data. Training

### Reference

1. Bray F, Ferlay J, Soerjomataram I, Siegel RL, Torre LA, Jemal A Global cancer statistics 2018: GLOBOCAN estimates of incidence and mortality worldwide for 36 cancers in 185 countries. CA: a cancer journal for clinicians. 2018 68(6):394-424.

2. Stankov Karmen. Bioinformatic tools for cancer geneticists. Archive of Oncology. 2015 13(2):69-75. https://doi.org/10.2298/AOO0502069S.

3. Campbell PJ, Getz G, Stuart JM, Korbel JO, Stein LD. Pan-cancer analysis of whole genomes. Nature. 2020 578:82-93. https://doi.org/10.1038/s41586-020-1969-6.

...

### Metadata

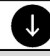

1. **Author**: Bray F, Ferlay J, Soerjomataram I, Siegel RL, Torre LA, Jemal A
**Title**: Global cancer statistics 2018: GLOBOCAN estimates of incidence and mortality worldwide for 36 cancers in 185 countries.
**Journal**: CA: a cancer journal for clinicians.
**Year**: 2018
**Volume**: 68
**Issue**: 6
**Page**: 394-424

...

**Fig 1. Example of bibliographic reference parsing.** In reference parsing, the input is a single reference string, and the output corresponds to metadata such as the author name, title, and journal. Parsed metadata field values are typically stored in machine-readable XML or JSON formats.

**Fig 2. Example of a sequence of labeled tokens for a reference string.** After tokenizing a reference string, each token is assigned specific labels. The tokens related to the metadata are assigned the labels for metadata field types. Other tokens not related to the metadata are assigned the "other" label.

data typically consist of a sequence of objects represented by the features of the references and a corresponding sequence of labels. To construct the training data, the reference string is transformed into a sequence of smaller and meaningful units, called tokens, using tokenization techniques [17–19]. After tokenization, each token is given a label that corresponds to one of the metadata field types, as Fig 2 shows. There are several ML algorithms for reference parsing, including hidden Markov models [20], support vector machines (SVM) [21, 22], and conditional random fields (CRF) [23–25].

Essentially, the performance of ML algorithms depends on the features that represent the training data. Existing reference parsers based on ML algorithms have achieved good performances using different handcrafted features. However, these features are dependent on specific domains; thus, they are not easily generalizable to other domains or reference styles. This problem can be overcome by using deep learning (DL) approaches that can automatically learn various representative features from the training data, and have a strong generalization performance. To the best of our knowledge, neural parscit [26] is the only DL-based tool that has been used to address the reference parsing problem. It employs a long short term memory (LSTM) to represent word embeddings and character-based word embeddings for reference strings. Afterward, the CRF model is used in a Softmax layer, yielding the final classification of the LSTM output.

The Korea Institute of Science and Technology Information (KISTI) is a government-funded research institute and data center in South Korea established to promote the efficiency of science and technology research and support high-tech research and development. Since early 2000, we have continuously collected various metadata including reference metadata from domestic and international scientific articles and national R&D reports. Collection of reference metadata, particularly from domestic articles, is performed manually. Therefore, a significant amount of money and time is required annually to build the database for storing reference metadata in domestic articles. There are two reasons why reference metadata have been manually extracted thus far. First, although several studies have suggested that the existing tools such as GROBID [24], CERMINE [25], and neural parscit [26] show good performance in extracting reference metadata, their accuracy is still insufficient to replace manual extraction. Second, domestic articles generally contain both English and non-English references. For the non-English references written in Korean, Chinese, Japanese, etc., the existing tools show inferior performance because they were developed based on English references.

Over the past few years, the field of natural language processing (NLP) has been rapidly transformed by an explosion in the use of neural networks and DL models [27]. Although many DL and transfer learning models, such as CNN, LSTM, Bi-LSTM, and BERT [28], can be easily modified and extended to various NLP problems, they are data-hungry, requiring large amounts of expensive labeled data. To the best of our knowledge, the building of a high-quality corpus is more important than developing a complex algorithm for extracting the

metadata from reference strings. Therefore, we focused more on constructing labeled data to enhance the automatic extraction of metadata from multilingual references. The contributions of this study are twofold:

1. We create a corpus covering multilingual journal article references. The corpus is annotated by domain experts with documental information and can be used to automatically extract reference metadata using DL models.

2. We conduct an experiment to demonstrate the effectiveness of our corpus. For this, we train and evaluate a BERT-based transfer learning model using our corpus. We compare the performance of the trained model with that of GROBID by entering completely new references not included in our corpus to each model.

We have made our corpus public on at http://doi.org/10.23057/47 to stimulate the development of text-mining systems for the automatic extraction of reference metadata. Text-mining systems trained using our corpus can be used by various institutions or companies that still rely on the manual extraction of reference metadata.

## Materials and methods

This section describes how we selected the candidate references and their corresponding metadata before starting manual annotation by professional annotators. Furthermore, we explain the details of the corpus construction based on the annotation guidelines. Finally, we explain how we trained and evaluated the BERT-based model using our corpus and how we compared the performance between GROBID and the BERT-based model. Fig 3 illustrates the entire process of building the corpus.

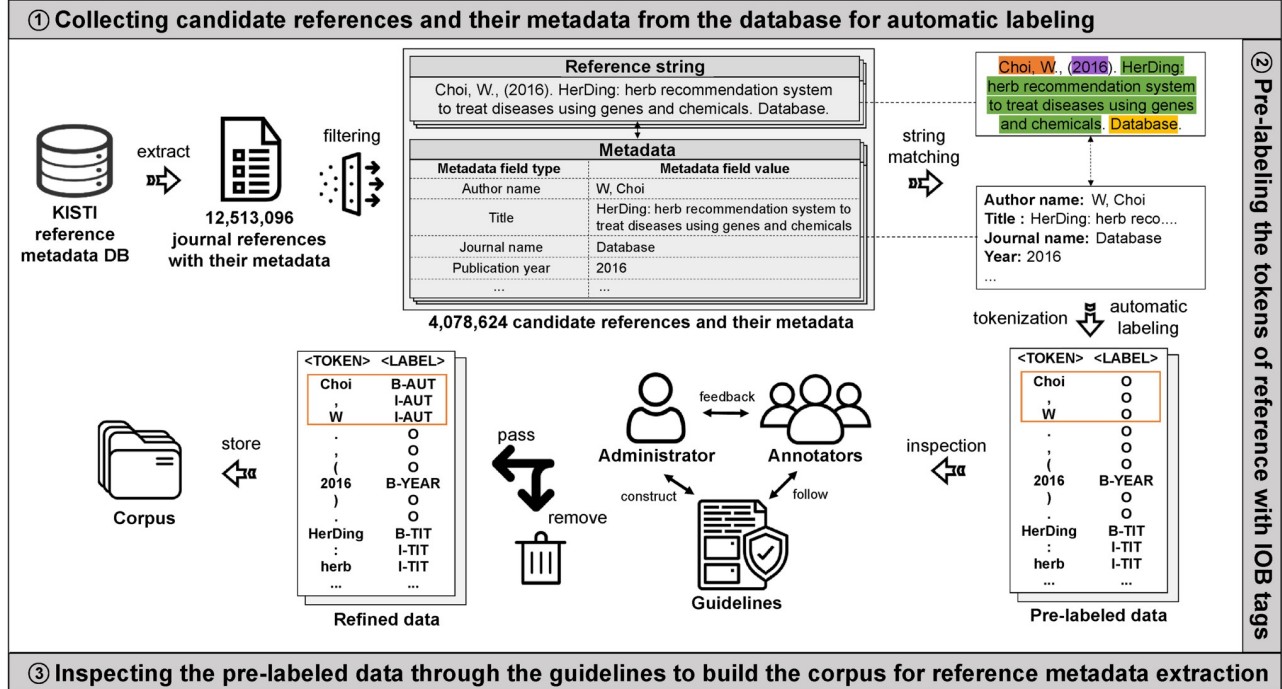

**Fig 3. Workflow of corpus construction.** The corpus was constructed as detailed in the following steps: (i) we first collected candidate references and their metadata from the KISTI database to automatically label the references; (ii) we then tokenized all the reference strings and pre-labeled the tokens of each reference string with IOB tags using a string matching approach based on the corresponding metadata field values; (iii) we precisely inspected whether the tokens of each reference string were correctly labeled based on the annotation guidelines.

## Collecting candidate references and their corresponding metadata

KISTI has been manually collecting reference metadata, especially on domestic journal articles and storing them in the database since early 2000. In the case of overseas publishers, they managed and stored the published international papers in refined forms such as XML and JSON, which can be further processed. Thus, high performance in extracting metadata field values from such international articles can be achieved with a rule-based approach. However, domestic journal articles published in South Korea such as those published by overseas publishers, have not been effectively managed. Therefore, KISTI has relied on manually extracting metadata field values from domestic articles to construct the metadata database. Every year, a significant amount of money and human resources are expended on building this database. Because the metadata were manually accumulated by curators over several decades, they can be now used to build a corpus for training and evaluating DL models that automatically extract the reference metadata. Therefore, in this study, we used existing metadata stored in our database to construct the corpus.

As of March 4, 2021, the database contains 17,032,742 multilingual references with corresponding metadata, as presented in Table 1. Among them, 12,513,096 references (73.46%) are journals and 11.57% and 4.91% are books and conferences, respectively. The remainder are theses, reports, and websites. In our study, to simplify the corpus construction and because the reference structure and metadata field types could be different for each reference type, we only used journal references. Domestic journal articles published in South Korea have cited many scientific papers from various countries. Thus, the references cited in the literature were typically written in various languages such as English, Korean, Chinese, and Japanese. Notably, English references constituted a large proportion.

To construct the corpus, we first extracted all the 12,513,096 journal references and their corresponding metadata from the database. From the extracted references, we removed the data that satisfied the following conditions: i) if a reference did not have transmission rights; ii) if an original reference string was missing; iii) if a reference contained duplicate values in different metadata fields (e.g., "2021" for year, "2021" for volume); iv) if a reference contained unnecessary characters such as HTML and LaTeX tags, and some broken characters; and v) if any of the metadata field values corresponding to the following three types of metadata (i.e., author name, title, journal name) was empty. We considered the data with the above-mentioned conditions as noise data that could be possibly problematic during corpus construction.

**Table 1. Number of references cited in domestic literature in our database as of March 4, 2021.**

| Reference type | Number of references (ratio) |
| --- | --- |
| Journal | 12,513,096 (73.46%) |
| Book | 1,971,968 (11.57%) |
| Conference | 836,577 (4.91%) |
| Thesis | 445,953 (2.62%) |
| Report | 345,073 (2.03%) |
| Website | 299,418 (1.76%) |
| Patent | 30,038 (0.18%) |
| Other | 590,619 (3.47%) |
| Total | 17,032,742 |

There are various reference types including journals, books, and conferences. The first and second columns present the reference type and the number of references stored in the database, respectively. The "other" refers to types other than those listed above, e.g., news articles.

| Reference string | Metadata fields | | | | | | | | | | | | |
|---|---|---|---|---|---|---|---|---|---|---|---|---|---|
| | Author name | Title | Journal name | Vol | Issue | Page | Year | DOI | URL | ISSN | Publisher | Pub_place | Pub_org |
| Spencer TE, Bazer FW. Uterine and placental factors regulating conceptus growth in domestic animals. J Anim Sci. 2004;82 Suppl 13:E4-13 https://.doi.org/10.2527/2004.8213_supplE4x | Spencer TE, Bazer FW | Uterine and placental factors regulating conceptus growth in domestic animals | J Anim Sci | 82 | Suppl 13 | E4-E13 | 2004 | 10.2527/2004.8213_supplE4x | http://dx.doi.org/10.2527/2004.8213_supplE4x | | | | |
| Park NR, Choi MS,Yang DH, Wu CH, Ahn HD. Chuna manual therapy for cervicogenic dizziness: a systematic review. The Journal of Korea CHUNA Manual Medicine for Spine & Nerves. 2018; 13(2):11-21 | Park NR, Choi MS, Yang DH, Wu CH, Ahn HD | Chuna manual therapy for cervicogenic dizziness: a systematic review | The Journal of Korea CHUNA Manual Medicine for Spine & Nerves | 13 | 2 | 11-21 | 2018 | | | | | | |
| A. A. Georgiev. Consistent nonparametric multiple regression: the fixed design case, J. Multivariate Anal. 25 (1988), no. 1, 100-110. https://doi.org/10.1016/0047-259X(88)90155-8 | A. A. Georgiev | Consistent nonparametric multiple regression: the fixed design case | J. Multivariate Anal. | 25 | 1 | 100-110 | 1988 | 10.1016/0047-259X(88)90155-8 | https://doi.org/10.1016/0047-259X(88)90155-8 | | | | |
| 황신해, 김민진 (2017). 보육교사의 CCTV인식 수준에 따른 직무스트레스와 전문성 인식. 유아교육연구, 37(1), 329-52 doi:10.18023/kjce.2017.37.1.014 | 황신해, 김민진 | 보육교사의 CCTV인식 수준에 따른 직무스트레스와 전문성 인식 | 유아교육연구 | 37 | 1 | 329-352 | 2017 | 10.18023/kjce.2017.37.1.014 | | | | | |
| 方璐瑶, '坚毅性与尽责性对大学生学业成绩的影响:刻意练习的中介作用--以浙江大学为例,' 高教论坛, 第5期, pp.103-9, 2019. | 方璐瑶 | 坚毅性与尽责性对大学生学业成绩的影响:刻意练习的中介作用--以浙江大学为例 | 高教论坛 | 5 | | 103-109 | 2019 | | | | | | |
| ... | ... | ... | ... | ... | ... | ... | ... | ... | ... | ... | ... | ... | ... |

**Fig 4. Example of references and their metadata field values extracted from the KISTI database.** The first column represents the reference strings. The rest of the columns describe the metadata field values, which were manually entered by curators. Depending on the ability of the curators, the quality of the entered values may vary.

After removing such data, we obtained 4,078,624 candidate references to be annotated during the corpus construction. Fig 4 shows an example of the candidate references and their corresponding metadata extracted from the database. Details of the corpus construction using candidate references are elaborated in the next subsection.

## Corpus construction process for the automatic extraction of metadata from multilingual references

This section describes how we built the corpus using a set of 4,078,624 candidate references and their metadata. Fig 3 summarizes the entire process of constructing the corpus. First, we pre-labeled the candidate references with IOB tags based on the CoNLL-2002 format [29] using metadata field values. As the labeled data with IOB tags where IOB referred to the inside, outside, and beginning of an entity were traditionally used as an encoding scheme for the named entity recognition task, they could be used for training and evaluating various DL models to automatically extract the reference metadata.

To pre-label the IOB tags, we first tokenized all the strings of the candidate references into separate tokens based on the following scheme: i) tokenize the reference string based on whitespaces; ii) tokenize the string based on the following special characters: !"()[]<>{}.,:;-_'$%&#?+*=@; and iii) do not tokenize the string corresponding to DOI or URL; rather it should be considered one token as it has a meaning in itself. After tokenization, we automatically assigned an appropriate IOB tag to each token via the exact string-matching approach as we already knew the metadata field values for each reference. For example, the pre-labeled reference in Fig 3 shows the result of automatically assigning IOB tags to the tokens of the reference string. The "B-TIT" tag was given to the "HerDing" token indicating the beginning of the article title, and the "I-TIT" label was given to the tokens corresponding to the remaining title tokens. Similarly, the "B-YEAR" label was given to "2016," representing the publication year, and the other tokens that were not related to the metadata were given the "O" tags. Through these IOB tags, the computer learned about the start and end of the tokens corresponding to each metadata, as well as the reference structures. Table 2 describes the 13 metadata field types.

As mentioned above, we automatically assigned IOB tags to candidate references based on the metadata field values extracted from the KISTI database. However, some of the metadata field values that were manually entered by human curators were inaccurate. For example, in

**Table 2. List of metadata field types in journal references.**

| Metadata field type | Description |
|---|---|
| AUT | Author name |
| TIT | Article title |
| JOU | Journal name |
| YEAR | Publication year |
| VOL | Volume |
| ISS | Issue number |
| PAGE | Page or article numbers |
| DOI | Digital object identifier |
| URL | Web address |
| ISSN | International standard serial number |
| PUBR | Publisher |
| PUB_PLC | Publication place |
| PUB_ORG | Publication organization |

Here, the aforementioned 13 types of metadata were considered. The article-sequence number or article number was used instead of the page range in the citation. As article numbers instead of page numbers are being increasingly used with the growing number of online journals, we considered the article numbers as the page numbers.

the pre-labeled reference as shown in Fig 3, the tokens corresponding to the author's name were incorrectly labeled with the "O" tags as the metadata field value for the author's name ("W, Choi") was different from the author's name represented in the reference string ("Choi, W"). Similarly, although the page numbers were represented as "329–52" in the original reference string as depicted in Fig 4, the actual metadata field value for the page was entered as "329–352." Therefore, it is possible that the tokens corresponding to the page were incorrectly labeled with the "O" tags or only a part of the page were correctly tagged because we used the exact string-matching approach. Therefore, an inspection process was deemed necessary.

During the manual inspection, primary and secondary inspections were performed on pre-labeled references. We hired eight professional annotators familiar with the literature for five months from April 28 to September 30, 2021, for the primary inspection. They were then grouped in pairs, and the annotators of each group carefully examined the same pre-labeled references. This was done to increase the reliability of the inspection results as much as possible and to subsequently compute the inter-annotator scores. To do this, an Excel file containing the pre-labeled references was provided to the annotators in each group, and each group examined a different portion of the data. Fig 5 shows a sample of this Excel file. Each reference in the file was identified by a unique reference identifier in the first line of the data. The tokens of each reference string and pre-labeled IOB tags were also represented. The annotators conducted the following steps with the given file:

1. First, the annotators carefully checked all the elements including tokens and IOB tags in the pre-labeled references. For example, it was necessary to check whether irrelevant IOB tags were assigned to any token of the reference string; whether tokenization was not properly done, or whether there was some problem with the reference itself.

2. Second, the annotators recorded their decisions in the last line of each reference in an Excel file. Essentially, "correct" was entered into the decision box if all the elements in the pre-labeled references were correct. Conversely, "incorrect" was entered if at least one element was incorrect. In some of the cases specified in the guidelines, both annotators in each group could manually correct the wrong parts after consultation with each other.

| REFID_OSSOB1_2020_v37n4_231_003 | |
|---|---|
| **TOKENS** | **LABELS** |
| Gould | B-AUT |
| <sp> | I-AUT |
| EN | I-AUT |
| , | I-AUT |
| <sp> | I-AUT |
| Cohen | I-AUT |
| … | … |
| … | … |
| <sp> | O |
| 18 | B-VOL |
| : | O |
| <sp> | O |
| 357 | B-PAGE |
| - | I-PAGE |
| 363 | I-PAGE |
| . | O |
| **ANNOTATOR DECISIONS** | |
| **Annotator 1** | **Annotator 2** |
| CORRECT | CORRECT |

| REFID_JBSHBC_2020_v27n4_3_040 | |
|---|---|
| **TOKENS** | **LABELS** |
| Gawin | B-AUT |
| , | I-AUT |
| <sp> | I-AUT |
| B | I-AUT |
| . | I-AUT |
| <sp> | I-AUT |
| … | … |
| … | … |
| , | O |
| <sp> | O |
| 347 | O |
| <sp> | O |
| - | O |
| <sp> | O |
| 358 | O |
| . | O |
| **ANNOTATOR DECISIONS** | |
| **Annotator 1** | **Annotator 2** |
| INCORRECT | INCORRECT |

**Fig 5. Sample of an Excel file used for inspection.** For each reference, the first column represents the token sequence. The second column shows the IOB tags that were automatically assigned by string matching. The annotators wrote their decisions in the decision box on the last line.

The administrator had defined the pre-labeled reference inspection criteria and constructed annotation guidelines in advance. After the primary inspections were completed, the administrator began the secondary inspections using the Excel file on which the primary inspection was completed. In the secondary inspection, the administrator checked for data with different decision-makings between the two annotators and informed the annotators of the correct decisions to gradually increase the consensus between the annotators. Furthermore, for the data that were manually corrected by the annotators, the administrator checked whether they were properly revised. If they were incorrect, the administrator corrected them. Detailed annotation guidelines for the aforementioned manual inspection process are described in the next section.

The manual inspection task was conducted for five months by eight annotators and the administrator. However, manually inspecting all 4,078,624 candidate references required a significant amount of time and labor. Owing to resource limitations, we used a programming language to automatically inspect the remaining candidate references. Based on the experience obtained from the manual inspection task, a rule-based program was developed. If all the elements in the remaining pre-labeled references were correct under the pre-defined conditions, they were automatically annotated as "correct" by the program, otherwise, they were annotated as "incorrect." Details of the conditions for decision-making in the inspection process are described in the next section.

## Annotation guidelines

Annotation guidelines for inspecting pre-labeled references were written by the administrator and updated periodically during the inspection process. The guidelines, which were distributed to the eight annotators during the manual inspection process are described in the "Manual annotation" section. Additionally, the criteria for automatically annotating the remaining pre-labeled references after the manual inspection task are explained in the "Fully automatic annotation" section. During the inspection, all the pre-labeled references were annotated as either "correct" or "incorrect." The definitions were as follows:

- Correct: if all the tokens and labels in the pre-labeled reference are correct.

- Incorrect: if one or more incorrect elements are found in the pre-labeled reference.

**Manual annotation.** During the primary inspection, annotators followed the following guidelines:

- The annotator had to check whether the tokens and their corresponding labels in a pre-labeled reference were all correct. If no error was found, the annotator entered "correct" in the decision box in the Excel file; otherwise, wrote "incorrect." For example, the annotator wrote "incorrect" when tokenization errors occurred.

- The annotator considered only the 13 metadata field types defined in Table 2 and ignored the other undefined types.

- Tokens that were unrelated to the metadata had to be labeled with "O" (non-metadata).

- A start token of metadata was labeled with "B" (beginning token), and the remaining tokens included in the scope of the corresponding metadata were labeled with "I" (inside token).

- "Et al." is short for the Latin term "et alia," meaning "and others." If this term appeared after the author's name in a reference string, the term was included in the scope of the author's name.

- Punctuation marks, such as commas or dots, separating the range of each metadata were labeled with "O."

- In some cases, some of the tokens corresponding to the author name were incorrectly placed after the title or journal name. In this case, the annotator entered "incorrect" in the decision box (e.g., *F. L. Teixeira and W. C. Chew*, *Advances in the theory of perfectly matched layers, in Fast and Efficient Algorithms in Computational Electromagnetics*, *W. C. Chew et al*., *eds., Artech House, Boston, 2001, pp. 283–346.*).

- Quotation marks placed at the front and end of the title were labelled with "O."

- Because article numbers are used as substitutes for page ranges in online journals, the article numbers were regarded as page numbers (e.g., *e04015014*).

- The annotator entered "incorrect" if the DOI and URL strings were tokenized into separate tokens.

- The tokens corresponding to URL access dates recorded after the URL were labeled with "O" (e.g., *[Accessed: March 10, 2020]*).

- If special characters such as HTML and LaTeX tags were included in a reference string, the annotator annotated it as "incorrect" in the decision box.

The annotators were not allowed to arbitrarily modify any content except the decision boxes in the Excel file. However, corrections were required for the cases described below. If the annotator found errors related to the conditions mentioned below, the modifications had to be confirmed by the administrator in advance. Thereafter, both the annotators of the same group simultaneously modified the corresponding contents in the same manner. The conditions for the corrections were as follows:

- If the scope of the author's names was incorrect, the annotator corrected the labels for the corresponding tokens. For example, in Fig 3, O tags assigned to the author's name, as described in the pre-labeled reference, were changed to "B-AUT" and "I-AUT."

- If the scope of the title was incorrect, the annotator corrected the labels for the corresponding tokens.

- If the scope of the journal name was incorrect, the annotator corrected the labels for the corresponding tokens.

- A digital object identifier (DOI) starting with "10." had a singular meaning. If the tokens corresponding to DOI were separated, they were combined into a single token and then labeled with "B-DOI."

- A URL has a meaning of its own. If the tokens corresponding to a URL were tokenized, they were combined into a single token and then labeled with "B-URL."

- The landing pages for the DOIs were labeled with "B-URL" instead of "B-DOI" (e.g., https://doi.org/10.1002/stc.384).

- The article numbers were mostly labeled with "O." Thus, they were changed to "B-PAGE" as the article numbers were considered as the page numbers.

- There were some cases in which the article and page numbers appeared together. In such cases, all the tokens corresponding to both the article and page numbers were labeled with the page tags (e.g., *102901(1)-102901(4)*).

- If the publication year was labeled with "O," the annotator had to modify it to "B-YEAR."

**Fully automatic annotation.** We automatically inspected the remaining candidate references after the manual inspection using a rule-based approach because of the limited availability of resources. In the pre-labeled references, where the following conditions were met, we assumed that there is a high possibility that errors were included. In these cases, we automatically annotated them as "incorrect." Conversely, the remaining references were annotated as "correct." The conditions for automatically annotating the pre-labeled references as "incorrect" were as follows:

- If either of B-AUT, B-TIT, or B-JOU did not exist in the pre-labeled references, they were regarded as errors.

- Usually there were no words before the author's name in the references. If any token before the token corresponding to "B-AUT" appeared in the pre-labeled references, they were regarded as errors.

- Any reference with a number or letter labeled with "O" between the author's name and the title was regarded as an error.

- If a DOI label was assigned to the token corresponding to a URL starting with "http," "https," or "www," it was regarded as an error.

- If an O tag was assigned to URL or DOI tokens, it was regarded as an error.

- If an "I-DOI" or "I-URL" tag existed, it was regarded as an error.

- Any number labeled with "O," it was regarded as an error.

- If "B-ISS" came before "B-VOL," it was regarded as an error.

- When the B-tag corresponding to each metadata field type appeared more than once, it was regarded as an error.

## Automatically extracting reference metadata using the DL model based on the annotated corpus

After the inspection, we trained and evaluated the DL models using our corpus to verify their reliabilities. BERT [28] is a transformer-based language model that is conceptually simple and empirically powerful. This model has been proven to have outstanding performance in various NLP tasks, and it is the first fine-tuning-based representation model that achieves state-of-the-art performance on several sentence-level and token-level tasks. In this experiment, we implemented this BERT-based transfer learning to automatically extract reference metadata and used the "Bert-base-multilingual-cased" version of the pre-trained model, which is trained on the top 104 languages with the largest Wikipedia because our corpus contained multilingual references. In this study, this BERT-based transfer-learning model is simply referred to as "*BERT-based model.*"

To train the BERT-based model, we used only the data annotated as "correct" among the automatically annotated references as the training dataset. Among the references, manually inspected by two annotators in each group, the references that both annotators decided to be correct were only used as validation and test datasets. Additionally, we created a new version of the datasets with whitespace tokens removed to evaluate the effects of whitespace tokens on performance. We employed a widely used Python implementation of BERT and Adam as the optimizer. For the hyperparameters, we empirically set the learning rate as [3e5, 5e5], number of epochs as [3, 4], batch size as 16 and max sequence length as 512. After training the models, we evaluated them on the test dataset and selected the state-of-the-art model.

GROBID is one of the most effective citation parsing tools, which uses CRF as the ML algorithm. Based on our experience, we believe that it is one of the most user-friendly software tools. It was first released in 2008 and has evolved over the years with new features. Moreover, it provides a GROBID service API to support a simple and efficient way to use it with Python or Java. According to [30], GROBID is still the best-performing tool to extract metadata from references. Therefore, we performed an experiment to show the superiority of the BERT-based model by comparing GROBID and the BERT-based model in terms of performance.

To this end, we first extracted 74,450 journal references with metadata field values that were registered in the KISTI database from July 29 to August 31, 2021. These references are new data that do not overlap with the references in our corpus. From these, we randomly collected 12,887 references with metadata field values. We periodically analyzed the patterns of the data errors in the KISTI database in terms of several assessment areas, including column consistency and uniqueness according to the database quality certification-value(DQC-V) of the Korea Data Agency, a government-affiliated organization aimed at building the circulation of a data ecosystem. In the analysis, the platinum class certification, which is granted only when the accuracy of data values in the database is 99.97% or higher, was obtained. Despite the

| Model | Reference string (input) | lang | Outputs | | | | | | | | | | | | | | | | | |
|---|---|---|---|---|---|---|---|---|---|---|---|---|---|---|---|---|---|---|---|---|
| | | | Author name | Decision | Title | Decision | Year | Decision | Journal name | Decision | Page | Decision | Vol | Decision | Issue | Decision | URL | Decision | DOI | Decision |
| GROBID | Oouchida Y, Sudo T, Inamura T, et al. Maladaptive change of body representation in the brain after damage to central or peripheral nervous system. Neurosci Res 2016;104:38-43. | eng | Oouchida, Y, Sudo, T, Inamura, T | Y | Maladaptive change of body representation in the brain after damage to central or peripheral nervous system | Y | 2016 | Y | Neurosci Res | Y | 38-43 | Y | 104 | Y | | Y | | Y | | Y |
| GROBID | Wang, Y., S. Chen, R.B. Blair, P. Jiang and P. Ding(2009) Nest composition adjustments by Chinese Bulbuls Pycnonotus sinensis in an urbanized landscape of Hangzhou (E China). Acta Ornithologica 44(2): 185-192. | eng | Wang, Y, Chen, S, Blair, R, Jiang, P, Ding, P | Y | Nest composition adjustments by Chinese Bulbuls Pycnonotus sinensis in an urbanized landscape of Hangzhou (E China) | Y | 2009 | Y | Acta Ornithologica | Y | 185-192 | Y | 44 | Y | 2 | Y | | Y | | Y |
| GROBID | 오훈근, 미디어 공간에서 사용자 경험의 인지적 작용 요소 연구, 한국공간디자인학회, 제4권 제3호, 2009 | kor | 오훈근, 미디어 공간에서 사용자 경험의 인지적 작용 요소, 연구, 한국공간디자인학회, 제3호 | N | | N | 2009 | Y | | N | | Y | | N | | N | | Y | | Y |
| GROBID | 長沼 光亮 (2000), '生物の生息環境としての日本海'. 日水研報告, 50, 1~42. | jpn | 長沼 光亮 | Y | | N | 2000 | Y | | N | 1-42 | Y | 50 | Y | | Y | | Y | | Y |

**Fig 6. Example of an Excel file containing the predicted metadata field values corresponding to the references newly extracted for comparing the performance of GROBID and the BERT-based model.** The first and second columns represent the model name and the input reference, respectively. The third column shows the language type of each reference string. The remainder are the predicted metadata field values with the decision boxes. The annotators manually entered their decisions into the decision boxes.

analysis, some metadata field values associated with the 12,887 references were incorrectly entered. Thus, the administrator manually corrected such incorrect values by scanning each value. After completing all these steps, we obtained completely reliable data. Henceforth, we refer to the 12,887 references and their metadata corrected by the administrator as "*new data*" and "*answer metadata*," respectively.

For the experiment, we used the new data as input to the two experimental models. For GROBID, we used the GROBID service API, which is based on Python. When a reference string is entered into this API, the metadata field values are returned in the text format. We also entered the reference strings into the BERT-based model to predict the metadata field values. Finally, the metadata field values extracted by each model were stored in the Excel file. The eight annotators manually checked whether the metadata field values extracted by each model were correct based on the given answer metadata. Fig 6 shows an example of the predicted metadata field values. There are decision boxes on the right next to each metadata field value. Thus, the annotators entered "Y" in a decision box if the corresponding value was correct; otherwise, they entered "N." For example, a specific metadata field type may not exist in the reference string. In this case, they entered "Y" if the predicted value is empty. If a predicted value in a specific metadata field type was unrelated to that type, they entered "N." Further, the details about the experimental results are described in the results section.

Unlike the aforedescribed manual verification method, we automatically calculated the string similarity between the answer metadata corresponding to the new data and the metadata predicted by the BERT-based model and GROBID using the Levenshtein distance, which is a performance metric used for determining the similarity between two strings– the source string (answer metadata) and the target string (predicted metadata by each model). Because the answer metadata were precisely verified by professional curators as aforementioned, the higher the similarity with the answer metadata, the better the model performance. Therefore, we analyzed the performance difference between the two models in predicting accurate metadata using this metric. The experimental results are detailed in the next section.

## Results

Table 3 presents the statistics of the number of references for each inspection step. Through inspection, we annotated 4,078,624 pre-labeled references. Among them, 144,934 were manually annotated by eight annotators in the primary inspection described in the materials and methods section. We selected 135,441 labeled references from the 144,934 annotated references by removing all but the pre-labeled references annotated as "correct" by both annotators in each annotation group. For references that were manually corrected by annotators during the primary inspection, the administrator manually checked whether they were properly revised. Errors, when found in the labeled data, were removed. During the secondary inspection, the administrator manually inspected 4,434 references revised by the annotators, and 74 were removed. Finally, we obtained 135,367 labeled references after all the above-mentioned steps were completed. They were later used as validation and test data for the BERT-based model. The remaining 3,933,690 pre-labeled references were automatically annotated as "correct" or "incorrect" according to the conditions described in the fully automatic annotation section. After automatic annotation, we obtained 3,680,620 labeled references that were annotated as "correct" and the remainder were removed. These were used as training data for the BERT-based model.

Consequently, our corpus currently contains 3,815,987 references labeled with the 13 metadata field types listed in Table 2. As our corpus was built to handle multilingual journal references, it covers various languages, such as English, Korean, Japanese, and Chinese. Table 4 describes the statistics for the number of references for each language. Many of the references cited in scientific articles published in the Republic of Korea are typically in English. Therefore, English references constituted the highest proportion of the references, followed by Korean, Japanese, and Chinese. As described in Fig 7, our corpus was in the form of a "txt" file

**Table 3. Statistics for the number of references by data type.**

| Data type | Number of references |
|---|---|
| Pre-labeled references | 4,078,624 |
| Candidates for the automatic inspection | 3,933,690 |
| Selected references after the automatic inspection | 3,680,620 |
| Candidates for the manual inspection | 144,934 |
| Selected references after the manual inspection | 135,367 |

**Table 4. Statistics for the number of references for each language in our corpus.**

| Language type | Number of references |
|---|---|
| English | 3,599,696 |
| Korean | 209,128 |
| Japanese | 3,028 |
| Chinese | 1,936 |
| German | 908 |
| French | 475 |
| Other | 816 |
| Total | 3,815,987 |

The corpus currently contains 3,815,987 labeled references and covers several languages including English, Korean, and Chinese. Most of references are written in English, followed by Korean, Japanese, and Chinese.

| ORBHBM_2020_v21n1_187_010 | English |
|---|---|
| Mendelson | B-AUT |
| , | I-AUT |
| <sp> | I-AUT |
| Nina | I-AUT |
| <sp> | I-AUT |
| A | I-AUT |
| . | O |
| <sp> | O |
| ' | O |
| Regulatory | B-TIT |
| <sp> | I-TIT |
| beneficiaries | I-TIT |
| <sp> | I-TIT |
| and | I-TIT |
| <sp> | I-TIT |
| informal | I-TIT |
| <sp> | I-TIT |
| agency | I-TIT |
| <sp> | I-TIT |
| policymaking | I-TIT |
| ; | O |
| <sp> | O |
| Cornell | B-JOU |
| <sp> | I-JOU |
| L | I-JOU |
| . | I-JOU |
| <sp> | I-JOU |
| Rev | I-JOU |
| . | O |
| <sp> | O |
| 92 | B-VOL |
| <sp> | O |
| ( | O |
| 2006 | B-YEAR |
| ) | O |
| . | O |
| \n | |

| TRMDBW_2020_v9_37_017 | Korean |
|---|---|
| 이정명 | B-AUT |
| , | O |
| <sp> | O |
| 심신통합적 | B-TIT |
| <sp> | I-TIT |
| 움직임 | I-TIT |
| <sp> | I-TIT |
| 교육으로서의 | I-TIT |
| <sp> | I-TIT |
| 무브먼트 | I-TIT |
| <sp> | I-TIT |
| 리츄얼 | I-TIT |
| ( | I-TIT |
| Movement | I-TIT |
| <sp> | I-TIT |
| Ritual | I-TIT |
| ) | I-TIT |
| 에 | I-TIT |
| <sp> | I-TIT |
| 관한 | I-TIT |
| <sp> | I-TIT |
| 고찰 | I-TIT |
| , | O |
| <sp> | O |
| 무용역사기록학 | B-JOU |
| <sp> | O |
| 제 | O |
| 32 | B-VOL |
| 집 | O |
| , | O |
| <sp> | O |
| 2014 | B-YEAR |
| . | O |
| \n | |

| KGOHCL_2021_v21n2_17_027 | German |
|---|---|
| Tae | B-AUT |
| - | I-AUT |
| Hwan | I-AUT |
| <sp> | I-AUT |
| Kim | I-AUT |
| , | O |
| <sp> | O |
| ' | O |
| Bemerkungen | B-TIT |
| <sp> | I-TIT |
| zur | I-TIT |
| <sp> | I-TIT |
| koreanischen | I-TIT |
| <sp> | I-TIT |
| Rezeption | I-TIT |
| <sp> | I-TIT |
| des | I-TIT |
| <sp> | I-TIT |
| Aktantenmodells | I-TIT |
| ' | O |
| , | O |
| <sp> | O |
| No | O |
| . | O |
| 10 | B-VOL |
| , | O |
| <sp> | O |
| pp | O |
| . | O |
| <sp> | O |
| 104 | B-PAGE |
| - | I-PAGE |
| 118 | I-PAGE |
| , | O |
| <sp> | O |
| 2001 | B-YEAR |
| . | O |
| \n | |

**Fig 7. Sample of the corpus used for extracting metadata from multilingual journal references.** The first column represents the tokens corresponding to the reference string and the second column represents the IOB tags corresponding to the tokens. Tokens and IOB labels were separated by tabs.

comprising two columns, where the first column represented the tokens corresponding to its reference string and the second column represented the IOB tags according to the corresponding tokens, and each reference was separated by a newline character. In the next section, we report on the calculation results of the inter-annotator agreement scores for annotating the 144,934 pre-labeled references. Furthermore, we explain our experimental results to demonstrate the effectiveness of the BERT-based model trained using our corpus.

## Inter-annotator agreement (IAA) score

As described, we manually annotated 144,934 pre-labeled references. Eight annotators took five months from April 28 to September 30, 2021 to complete this process. To improve the corpus quality, the annotators were grouped in pairs, and each annotator of a group annotated the same pre-labeled references. Inter-annotator agreement is used to measure the consistency in the annotation of a particular class of the given data between two annotators. Therefore, to demonstrate the reliability of the annotated results, we measured the inter-annotator agreement scores using Cohen's kappa statistic [31], which is the most frequently used method for measuring the overall agreement between two annotators. According to [32], Cohen's kappa values were interpreted as follows: 0.81–0.99 denotes an almost perfect agreement, 0.61–0.8 denotes a substantial agreement, 0.41–0.6 denotes a moderate agreement, 0.21–0.4 denotes a fair agreement, and $\leq 0.20$ denotes a poor agreement.

We used $2 \times 2$ contingency tables to evaluate the annotation results of each pair of annotators (see Table 5). Note that the term *correct* indicates that the annotator determined that there were no errors in the given data, and the term *incorrect* indicates that the annotator determined that there were errors in the given data. The components of the kappa value were

**Table 5. 2 × 2 Contingency table for two annotators.**

| | | Annotator β | | |
|---|---|---|---|---|
| | | **Correct** | **Incorrect** | |
| Annotator α | Correct | *A* | *B* | *A+B* |
| | Incorrect | *C* | *D* | *C+D* |
| | | *A+C* | *B+D* | *N(A+B+C+D)* |

defined as follows: $p_0$, the proportion of units in which there is an agreement (observed accuracy) is

$$p_0 = \frac{A + D}{N}. \tag{1}$$

$p_e$, the proportion of units in which agreement is expected by chance (theoretical accuracy) is

$$p_e = \frac{A + B}{N} \times \frac{A + C}{N} + \frac{B + D}{N} \times \frac{C + D}{N}. \tag{2}$$

The Cohen's kappa value ($\kappa$) is calculated as follows:

$$\kappa = \frac{p_0 - p_e}{1 - p_e}. \tag{3}$$

Table 6 presents the annotation results for the given pre-labeled references for each group. Based on these annotation results, we calculated the kappa values, as described in Table 7. The kappa value for Group 3 was the highest, at 0.941, while that for Group 1 was the lowest, at 0.84. The annotators in group 1 were hired first, and they started to construct their annotations, whereas the remaining annotators were hired two months later. We believe that the

**Table 6. Annotation results for Group 1 to 4.**

| Group 1 | | Annotator 2 | | |
|---|---|---|---|---|
| | | Correct | Incorrect | |
| Annotator 1 | Correct | 34,103 | 166 | 34,269 |
| | Incorrect | 382 | 1,547 | 1,929 |
| | | 34,485 | 1,713 | 36,198 |
| Group 2 | | Annotator 4 | | |
| | | Correct | Incorrect | |
| Annotator 3 | Correct | 33,825 | 126 | 33,951 |
| | Incorrect | 226 | 2,037 | 2,263 |
| | | 34,051 | 2,163 | 36,214 |
| Group 3 | | Annotator 6 | | |
| | | Correct | Incorrect | |
| Annotator 5 | Correct | 32,399 | 149 | 32,548 |
| | Incorrect | 102 | 2,163 | 2,265 |
| | | 32,501 | 2,312 | 34,813 |
| Group 4 | | Annotator 8 | | |
| | | Correct | Incorrect | |
| Annotator 7 | Correct | 35,114 | 95 | 35,209 |
| | Incorrect | 343 | 2,157 | 2,500 |
| | | 35,457 | 2,252 | 37,709 |

**Table 7. Cohen's kappa values for the annotation results of each group.**

| Group number | N | $p_o$ | $p_e$ | kappa ($\kappa$) |
|---|---|---|---|---|
| Group 1 | 36,198 | 0.98486 | 0.90496 | 0.842 |
| Group 2 | 36,214 | 0.99028 | 0.88524 | 0.915 |
| Group 3 | 34,813 | 0.99279 | 0.87717 | 0.941 |
| Group 4 | 37,709 | 0.98839 | 0.8819 | 0.902 |
| All | 144,934 | 0.98904 | 0.88716 | 0.903 |

agreement rates between the numbers of group 1 were relatively lower than those of group 2, because several trials and errors were conducted at the beginning of the corpus annotation process. Whereas, the remaining members had relatively higher agreement rates because improved guidelines at the beginning of the annotation process were provided. Consequently, we annotated the total 144,934 pre-labeled references and achieved an overall IAA score of 0.903, which is considered "*almost perfect*" agreement according to [32].

## Training and evaluating BERT-based transfer learning models using our corpus

To verify the effectiveness of the constructed corpus, we trained and evaluated the performance of BERT-based transfer-learning models for extracting reference metadata based on our corpus. For this, we used 3,680,620 labeled references constructed through the automatic inspection as the training set. Of the 135,367 manually inspected references, 63,878 were used as the validation set, and 71,489 as the test set. These data contain whitespace tokens denoted as "<sp>." To determine whether these tokens affect the prediction performance, we prepared identical datasets without whitespace tokens. Then, we trained and evaluated the BERT-based models according to the hyper-parameters previously mentioned in the methods section and the two types of datasets. For transfer learning, we used the "Bert-base-multilingual-cased" pre-trained model. The BERT-based models were implemented using Python and based on the BERT source code. Our experiments were conducted on a workstation with an Intel(R) Xeon(R) Gold 6226 CPU, 125 GB RAM, and two Tesla V100 32GB GPUs. As we set the max sequence length to 512, the learning time was quite long; it took approximately 45.71 hours per epoch. The trained models predict the IOB labels that are related to the metadata field types for the given input tokens of references. Therefore, we evaluated whether the predicted IOB labels for tokens in the test set were correct. Table 8 presents the evaluation results. The

**Table 8. Model performances according to hyper-parameters for transfer learning and presence or absence of whitespace tokens in the corpus.**

| Models | <sp> tokens? | epoch | learning_rate | P(%) | R(%) | F1(%) |
|---|---|---|---|---|---|---|
| BERT-REF1 | included | 3 | 3e5 | 99.79 | 99.87 | 99.83 |
| BERT-REF2 | included | 3 | 5e5 | 99.78 | 99.85 | 99.82 |
| BERT-REF3 | included | 4 | 3e5 | 99.72 | 99.81 | 99.77 |
| BERT-REF4 | included | 4 | 5e5 | 99.78 | 99.85 | 99.82 |
| BERT-REF5 | removed | 3 | 3e5 | 99.77 | 99.85 | 99.81 |
| BERT-REF6 | removed | 3 | 5e5 | 99.69 | 99.79 | 99.74 |
| BERT-REF7 | removed | 4 | 3e5 | 99.7 | 99.74 | 99.72 |
| BERT-REF8 | removed | 4 | 5e5 | 99.68 | 99.79 | 99.74 |

The batch size and maximum sequence length were fixed at 16 and 512, respectively.

state-of-the-art model, BERT-REF1, achieved an F1-score of 99.83%, and other models showed performances similar to BERT-REF1. As presented in Table 8, when whitespace tokens were removed from the corpus, the performance was slightly reduced.

## Performance comparison of the BERT-based model and GROBID using new references

As described in the previous section, we obtained the BERT-REF1 model through BERT-based transfer learning using our corpus to extract the reference metadata. To compare BERT-REF1 with GROBID in performance, we prepared 12,887 new data and their answer metadata, and new data were entered as inputs for the two models. Finally, each model predicted the metadata field values corresponding to each reference in the given new data. As shown in Fig 6, the predicted metadata field values were stored in an Excel file. The data in the file were divided and distributed evenly among the annotators. Based on the answer metadata constructed by the administrator, the annotators checked whether the given metadata field values were correctly predicted by each model. If the metadata field value and its range were correct, the annotator entered "Y;" otherwise, they entered "N" in the decision boxes.

Based on the evaluations by these annotators, the accuracy of the predicted values for each metadata field type was calculated using the number of "Y" decisions in each metadata field type divided by the total number of new data (= 12,887). Table 9 presents the number of metadata field values for each field type in the new data according to the answer metadata. The lower six types, including URL and DOI, are relatively few in the new data compared to the other types. Thus, the accuracy for these six types was inevitably high because both models typically returned empty strings when they were not included in the reference strings. For example, most of the values in the decision boxes for ISSN were "Y" because ISSN numbers rarely appeared in the new data. Therefore, in this experiment, we only focused on the accuracy of the top 7 types.

Table 10 shows the accuracy of the metadata field values for each type extracted from the new data using BERT-REF1 and GROBID. For all the metadata field types, BERT-REF1 showed significantly higher accuracy than GROBID. BERT-REF1 achieved an average accuracy of 99.62%, which was 15.15% higher than that of GROBID. The accuracies of the author's name, title, and journal name were relatively low in GROBID. Furthermore, we separately compared the accuracy of the metadata field values extracted from English and non-English

**Table 9. Number of metadata field values in 12,887 new references.**

| Metadata field type | Number of metadata field values |
|---|---|
| Journal name | 12,887 |
| Author name | 12,877 |
| Publication year | 12,877 |
| Title | 12,805 |
| Volume | 12,502 |
| Page | 12,219 |
| Issue number | 7,294 |
| DOI | 2,033 |
| URL | 1,502 |
| Publication place | 96 |
| Publisher | 8 |
| Publication organization | 4 |
| ISSN | 3 |

**Table 10. Accuracy of the metadata field values extracted from 12,887 new references using BERT-REF1 and GROBID.**

| Metadata field type | GROBID (%) | BERT-REF1(%) |
| --- | --- | --- |
| Author name | 77.64 | 99.85 |
| Title | 70.77 | 99.7 |
| Journal name | 71.3 | 99.49 |
| Publication year | 97.34 | 99.94 |
| Volume | 89.01 | 99.24 |
| Issue number | 87.74 | 99.6 |
| Page number | 97.46 | 99.51 |
| Avg. | 84.47 | 99.62 |

references as our corpus contains references in various languages, particularly Korean. In the new data, 9,539 and 3,348 were English and non-English references, respectively. Table 11 shows the accuracy of metadata field values for each type extracted from English references. The accuracy of BERT-REF1 was slightly higher by 2.72%; GROBID also showed good performance for English references. The accuracy of metadata field values for each type extracted from non-English references are shown in Table 12. Contrary to the English reference results, GROBID showed relatively poor performance for all other types except for publication years and page numbers. In comparison, BERT-REF1 still achieved an average accuracy of 99.2%, which is a 50.58% improvement over GROBID.

As explained above, BERT-REF1 exhibited excellent performance in extracting reference metadata. We analyzed several cases in which the prediction results for some references were

**Table 11. Accuracy of the metadata field values extracted from 9,539 English references using BERT-REF1 and GROBID.**

| Metadata field type | GROBID (%) | BERT-REF1(%) |
| --- | --- | --- |
| Author name | 99.27 | 99.84 |
| Title | 94.62 | 99.69 |
| Journal name | 92.2 | 99.58 |
| Publication year | 99.13 | 99.93 |
| Volume | 98.2 | 99.67 |
| Issue number | 98.46 | 99.82 |
| Page number | 97.47 | 99.83 |
| Avg. | 97.05 | 99.77 |

**Table 12. Accuracy of the metadata field values extracted from 3,348 non-English references using BERT-REF1 and GROBID.**

| Metadata field type | GROBID (%) | BERT-REF1(%) |
| --- | --- | --- |
| Author name | 16.04 | 99.88 |
| Title | 2.81 | 99.73 |
| Journal name | 11.77 | 99.22 |
| Publication year | 92.23 | 99.97 |
| Volume | 62.84 | 98.03 |
| Issue number | 57.2 | 98.99 |
| Page number | 97.43 | 98.6 |
| Avg. | 48.62 | 99.2 |

inaccurate. This problem can be addressed by applying a rule-based method in the future. The reasons for the inaccurate predictions are as follows:

- Although it is rare for the sequence length of tokens to exceed 512, BERT-REF1 cannot extract reference metadata if the sequence length of the reference string exceeds 512 owing to too many author's names.

- When the tokens related to journal name and publication year were not separated by whitespaces owing to typos, the metadata field values for journal name and year could not be properly extracted.

  - *Malav, A., Kadam, K., Kamat, P.. Prediction of heart disease using k-means and artificial neural network as hybrid approach to improve accuracy. International Journal of Engineering and* **Technology2017***;9(4):3081–3085.*

- If there is no page number in the reference string and the month and date are right next to the journal name, the date could be misidentified as the page number.

  - *Yang, H., Kuo, Y.H., Smith, Z.I., and Spangler, J. (2021). Targeting cancer metastasis with antibody therapeutics. Wiley Interdisc. Rev. Nanomed. Nanobiotechnol. 2021 Jan* **18** *[Epub].* https://doi.org/10.1002/wnan.1698

- If there is no volume number in the reference string and the month and date are right next to the journal name, the date could be misidentified as the volume number.

  - *Z. Akhtar, J. W. Lee, M. A. Khan, M. Sharif, S. A. Khan, and N. Riaz, "Optical character recognition (OCR) using partial least square (PLS) based feature reduction: an application to artificial intelligence for biometric identification," Journal of Enterprise Information Management, Jul.* **31** *2020.*

- If the title is missing from the reference string for some reason, other metadata at the location where the title should be, for example, the journal name, could be incorrectly predicted as the title.

  - *Tsutsumi, T.; Akiyama, H.; Demizu, Y.; Uchiyama, N.; Masada, S.; Tsuji, G.; Arai, R.; Abe, Y.; Hakamatsuka, T.; Izutsu, K.; Goda, Y.; Okuda, H.* **Biol. Pharm. Bull***. 2019, 42, 547, DOI*: 10.1248/bpb.b19-00006.

Lastly, we calculated the string similarity between the answer metadata and the metadata predicted by BERT-REF1 and GROBID using Levenshtein distance. Fig 8 depicts the similarity

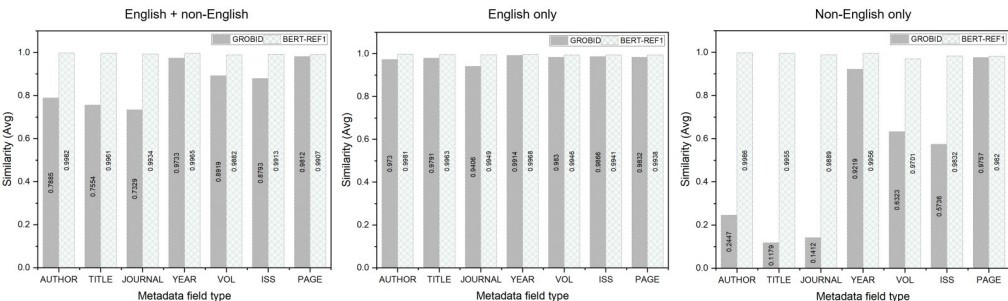

**Fig 8. Similarity performance of GROBID and BERT-REF1 for the metadata field types based on Levenshtein distance.** The first graph shows the results for all the 12,887 references; the rest represent the results for the 9,539 English and 3,348 non-English references.

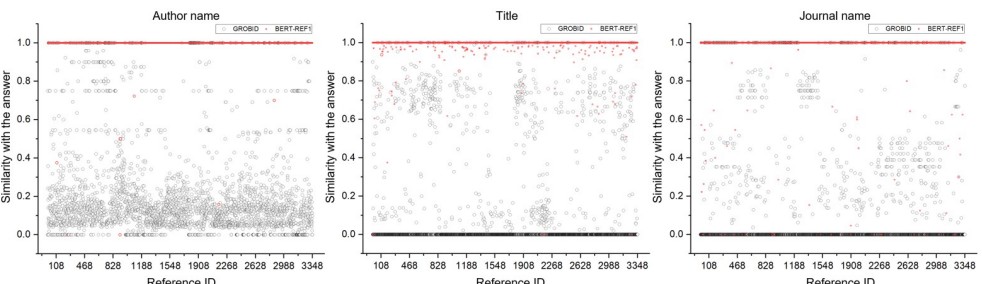

**Fig 9. Visualization of the similarity values and differences between BERT-REF1 and GROBID for the three metadata field types in the 3,348 non-English references.** The X-axis represents each non-English reference. The red circles indicate the similarity values for the non-English references for BERT-REF1, and the black circles represent the corresponding similarity values for GROBID.

scores according to the metadata field types. Notably, the closer the similarity is to 1, the higher is the string similarity with the answer metadata. In Fig 8, the first graph represents the results for all the 12,887 references, and the others illustrate the results for the 9,539 English references and 3,348 non-English references. Similar to the manual verification results shown in Table 11, BERT-REF1 slightly outperformed GROBID in terms of similarity for the English references. However, for the non-English references, BERT-REF1 significantly outperformed GROBID. It was found that the similarity performances of the two models in terms of the author's names, titles, and journal names were significantly different. For example, for BERT-REF1, the similarities in terms of the author's name, title, and journal name were higher than those of GROBID by 0.7539, 0.8776, and 0.8477, respectively. The Excel sheets containing the answer metadata and metadata predicted by BERT-REF1 and GROBID for all the 12,887 references are provided as supplementary information in S1 File.

Additionally, we visualized the similarity between BERT-REF1 and GROBID for the three metadata field types in the 3,348 non-English references as shown in Fig 9. As can be seen, the red circles indicate the similarity values for the non-English references for BERT-REF1, and the black circles represent the corresponding similarity values for GROBID. For the three metadata field types, most of the red circles, representing the similarity of metadata predicted by BERT-REF1 are distributed close to 1.0. However, the black circles, indicating the similarity of metadata predicted by GROBID, are likely to be distributed close to 0.0. This implies that GROBID could not precisely extract the metadata from the given non-English references. Therefore, we conclude that BERT-REF1 outperformed GROBID in extracting the metadata from references.

## Conclusion

This study presented a detailed description of our procedure to construct a corpus for extracting multilingual reference metadata. The corpus contains 3,815,987 references labeled with IOB tags corresponding to 13 metadata field types. Among them, 135,367 were manually labeled by annotators and an administrator, and 3,680,620 were labeled using an automated process. Because we focused more on constructing a corpus that can be used to extract metadata from multilingual references, we included English references as well as those in other languages, particularly Korean. Through experiments, we demonstrated the reliability of our corpus by comparing the performances of BERT-REF1 and GROBID. Our corpus contributed to excellent performances when extracting the metadata from references written in English as well as those written in other languages. Therefore, the generated corpus could serve as a gold standard for developing tools for extracting metadata from multilingual references. However,

the developed corpus has some limitations that need to be addressed in the future. Because the corpus provides only journal-type references, the performance in the case of other reference types such as conferences and books could be relatively less desirable. Moreover, as shown in Table 4, most of the references in the corpus are in English and Korean. Therefore, the ratio of references by language needs to be considered in future work. Further, we intend to reinforce the following to improve the coverage of the corpus: (i) We will expand the corpus to other reference types. (ii) We will add additional references in other languages. (iii) We will add a rule-based approach to solve some prediction errors of BERT-REF1.

## Supporting information

**S1 File. The answer metadata and the metadata predicted by BERT-REF1 and GROBID for all the 12,887 references to measure the similarity performances of the two models based on the Levenshtein distance.**
(XLSX)

## Author Contributions

**Data curation:** Wonjun Choi.

**Formal analysis:** Wonjun Choi.

**Funding acquisition:** Hwa-Mook Yoon, Mi-Hwan Hyun, Hye-Jin Lee.

**Investigation:** Wonjun Choi, Kangsan Dajeong Lee.

**Methodology:** Wonjun Choi.

**Project administration:** Hyesoo Kong.

**Resources:** Wonjun Choi, Jae-Wook Seol, Young Joon Yoon, Hyesoo Kong.

**Software:** Wonjun Choi, Hyesoo Kong.

**Supervision:** Hwa-Mook Yoon, Mi-Hwan Hyun.

**Validation:** Wonjun Choi.

**Visualization:** Wonjun Choi.

**Writing – original draft:** Wonjun Choi.

**Writing – review & editing:** Wonjun Choi, Hyesoo Kong.

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
