## [Decision Letter · Decision Letter 0]

31 Aug 2022

PONE-D-22-06759Building an annotated corpus for automatic metadata extraction from multilingual journal article referencesPLOS ONE

Dear Dr. Kong,

Thank you for submitting your manuscript to PLOS ONE. After careful consideration, we feel that it has merit but does not fully meet PLOS ONE’s publication criteria as it currently stands. Therefore, we invite you to submit a revised version of the manuscript that addresses the points raised during the review process.

Please see the reviewers' comments below. We ask that you ensure you address each of the reviewers' comments when revising your manuscript.

We look forward to receiving your revised manuscript.

Kind regards,

Hugh Cowley

Staff Editor

PLOS ONE

Journal Requirements:

“This research was supported by the Korea Institue of Science and Technology Information (KISTI) of the Ministry of Science and ICT, South Korea (MSIT) (No. K-22-L01-C01-S01: Construction on Intelligent SciTech Information Curation). The funders had no role in study design, data collection and analysis, decision to publish, or preparation of the manuscript.”

Reviewers' comments:

Reviewer's Responses to Questions

**Comments to the Author**

1. Is the manuscript technically sound, and do the data support the conclusions?

Reviewer #1: Yes

Reviewer #2: Yes

2. Has the statistical analysis been performed appropriately and rigorously? 

Reviewer #1: Yes

Reviewer #2: N/A

3. Have the authors made all data underlying the findings in their manuscript fully available?

Reviewer #1: Yes

Reviewer #2: Yes

4. Is the manuscript presented in an intelligible fashion and written in standard English?

Reviewer #1: Yes

Reviewer #2: Yes

5. Review Comments to the Author

Reviewer #1: This work describes the use of data from a large database of manually annotated article references to create a corpus of reference strings and tagged token sequences from those reference strings which represent correct parses of the reference strings. The tags used are of the IOB type and are made specific to the thirteen field types studied. At this point the references are limited to references found in journal articles. Currently the corpus contains 3,815,987 multilingual references. While most references are in English there are a significant number in Korean, Chinese and Japanese. A few are in German or French. The creation of this corpus began with 4,078,624 references which were tokenized and automatically tagged based on the metadata contained in the original database. A set of 144,934 of these were selected for manual review by eight annotators working in pairs. The pairs of annotators had a high level of agreement in deciding if references were correctly parsed (Cohen’s Kappa average of 0.903). From this work 135,367 references were either correctly parsed or could be corrected to be correctly parsed and thus usable and correct to a human standard. The remaining references were subjected to an automatic screening process and 3,680,620 were judged correct based on this screening. These were used to train a BERT model to automatically tag tokenized reference strings. The 135,367 references were divided into validation and tests and the trained BERT model predicted the test data with an F1 of 99.83. The BERT model was compared with the GROBID algorithm for the same task and BERT proved to be much better.

The paper is well written in good English and details are clearly explained.

Suggestions: 1) Is locating the references strings in journal articles a problem? If so, is your data helpful in solving it?

2) You state that previous methods of parsing reference strings do not perform sufficiently well to be used without human intervention. Does your BERT model produce results good enough to be used without a human examining results? Are you using it in your curation process for your KISTI database?

3) p. 12, line 183 “each group delicately examined” would be better “each group carefully examined.”

Reviewer #2: The paper presents a procedure to build an annotated corpus of bibliographic references using

metadata. The authors proposed reference labeling using pre-trained BERT models.

The topic is relevant, and the article explores promising ideas in the context of the annotated corpus.

All assessment scenarios have better results (Tab 10, 11, and 12). In this sense, doubts about

the work results can be pointed out. One suggestion is to propose an extra analysis to validate

the results through textual data visualization techniques. For example, agglomerative clustering

methods can be proposed to visualize the results.

6. PLOS authors have the option to publish the peer review history of their article (what does this mean?). If published, this will include your full peer review and any attached files.

Reviewer #1: No

Reviewer #2: No

---

## [Author Response · Author response to Decision Letter 0]

10 Oct 2022

Reviewer#1, comment #1: Is locating the references strings in journal articles a problem? If so, is your data helpful in solving it?

[Author response] Thank you for the comment. We have collected, stored and managed all journal articles in PDF format. Therefore, it is necessary to find and extract the list of references included in the PDF before the metadata extraction process starts. We have been collecting journal articles from over 500 domestic journals annually. Because each journal has a different design of the paper and reference numbering style, it is difficult to automatically locate and extract the list of references in journal articles. Therefore, we have manually extracted them thus far. The corpus mentioned in this paper is built to extract metadata automatically under the assumption that a single reference is given. Recently, we started a study aimed extracting the bibliography list from pdf papers automatically. In more detail, we plan to construct an image-based corpus for detecting the locations of references and for separating into individual references. Using this corpus, we will train the object detection models such as YOLO. For extracting reference texts from the detected objects, we will use the OCR technologies. In the near future, we plan to fully automate the bibliographic metadata extraction process.

Reviewer#1, comment #2: You state that previous methods of parsing reference strings do not perform sufficiently well to be used without human intervention. Does your BERT model produce results good enough to be used without a human examining results? Are you using it in your curation process for your KISTI database?

[Author response] Thank you for the pertinent comment. The journal articles collected by KISTI include many non-English references and English. For these non-English references, existing methods did not correctly extract their metadata. On the other hand, as mentioned in the paper, the metadata extracted by BERT-REF1 were almost identical to the strings of answer metadata verified by humans. In addition, after we conducted this research, we continuously accumulated the results of both the manual extraction of metadata and the automatic extraction using BERT-REF1 for several months. In conclusion, it was found that BERT-REF1 produced sufficiently good results to be used without human intervention. Currently, the metadata for given references are automatically extracted without any human intervention and accumulated in the KISTI DB. Therefore, BERT-REF1 is now applied to fully automate the metadata extraction of references during the curation process for the KISTI Database.

Reviewer#1, comment #3: p.12, line 183 “each group delicately examined” would be better “each group carefully examined.”

[Author response] Thank you for the comment. We have corrected the term accordingly. We changed the term "delicately" to "carefully."

Reviewer#2, comment #1: All assessment scenarios have better results (Tab 10, 11, and 12). In this sense, doubts about the work results can be pointed out. One suggestion is to propose an extra analysis to validate the results through textual data visualization techniques. For example, agglomerative clustering methods can be proposed to visualize the results.

[Author response] Thank you for the suggestion. Tables 10, 11, and 12 show the accuracies calculated by the annotators manually comparing the answer metadata with the metadata predicted by GROBID and BERT-REF1. Therefore, questions concerning the the results can arise because they result from manual evaluation by annotators. Therefore, we attempted to perform an additional analysis to validate the results through textual data visualization techniques such as agglomerative clustering. However, we found it challenging to visualize the validation results using clustering methods because our experiment, as shown in Tables 10, 11, and 12, simply aimed to verify equality between the answer metadata and the predicted metadata by each model. Therefore, we performed another experiment that automatically compares the string similarities between the answer metadata and the predicted metadata by each model. 

In the experiment, we automatically calculated the string similarities between the answer metadata and the metadata predicted by BERT-REF1 and GROBID using the Levenshtein distance. The answer metadata for the 12,887 references mentioned in the paper can be considered accurate and reliable since the KISTI database has been regularly evaluated by the Korea Data Agency, a government-affiliated organization, to check the DB quality, and the administrator of this research additionally inspected them. Thus, it can be considered that the higher the similarity with the answer metadata is, the better the model performance. Figure 8 shows the similarities according to the metadata field types. Similar to the manual evaluation results shown in Tables 10, 11, and 12, BERT-REF1 slightly outperformed the similarity performance of GROBID for the English references. However, for non-English references, BERT-REF1 showed significantly better performance than GROBID. In particular, it was found that the similarity performance between the two models for author names, titles, and journal names in non-English references was significantly different. We also visualized the similarity difference between BERT-REF1 and GROBID for the above three metadata field types in 3,348 non-English references, as shown in Figure 9. In Figure 9, the red similarity points of metadata predicted by BERT-REF1 were distributed closer to 1.0 than GROBID. This is because GROBID could not extract accurate metadata from non-English references. Based on this experiment, we concluded that BERT-REF1 outperformed GROBID not only in English references but also in non-English references. We also added Excel sheets containing the answer metadata and the metadata predicted by BERT-REF1 and GROBID for all 12,887 references as supplementary information.

---

## [Decision Letter · Decision Letter 1]

6 Dec 2022

PONE-D-22-06759R1Building an annotated corpus for automatic metadata extraction from multilingual journal article referencesPLOS ONE

Dear Dr. Kong,

Thank you for submitting your manuscript to PLOS ONE. After careful consideration, we feel that it has merit but does not fully meet PLOS ONE’s publication criteria as it currently stands. Therefore, we invite you to submit a revised version of the manuscript that addresses the points raised during the review process.

The paper needs to be proofread for English enhancements.

We look forward to receiving your revised manuscript.

Kind regards,

Sanaa Kaddoura

Academic Editor

PLOS ONE

Journal Requirements:

Reviewers' comments:

Reviewer's Responses to Questions

**Comments to the Author**

1. If the authors have adequately addressed your comments raised in a previous round of review and you feel that this manuscript is now acceptable for publication, you may indicate that here to bypass the “Comments to the Author” section, enter your conflict of interest statement in the “Confidential to Editor” section, and submit your "Accept" recommendation.

Reviewer #1: All comments have been addressed

Reviewer #3: All comments have been addressed

2. Is the manuscript technically sound, and do the data support the conclusions?

Reviewer #1: (No Response)

Reviewer #3: Yes

3. Has the statistical analysis been performed appropriately and rigorously? 

Reviewer #1: (No Response)

Reviewer #3: Yes

4. Have the authors made all data underlying the findings in their manuscript fully available?

Reviewer #1: (No Response)

Reviewer #3: No

5. Is the manuscript presented in an intelligible fashion and written in standard English?

Reviewer #1: (No Response)

Reviewer #3: Yes

6. Review Comments to the Author

Reviewer #1: (No Response)

Reviewer #3: the author (s) improved the paper based on the received comments. the paper can be published after improving language.

7. PLOS authors have the option to publish the peer review history of their article (what does this mean?). If published, this will include your full peer review and any attached files.

Reviewer #1: No

Reviewer #3: No

---

## [Author Response · Author response to Decision Letter 1]

2 Jan 2023

[Comments from Editor]

Author response: In response to the above comments, we reviewed our reference list to ensure that it is complete and correct. 

Author action: We updated some of the references as follows:

(reference #3, page 19) Pranckut˙e, R Web of Science (WoS) and Scopus: The titans of bibliographic information in today’s academic world. Publications. 2021; 9(1):12. https://doi.org/10.3390/publications9010012

(reference #5, page 19) Khabsa M, Giles CL. The Number of Scholarly Documents on the Public Web. PLoS ONE. 2014; 9(5):e93949. https://doi.org/10.1371/journal.pone.0093949

(reference #18, page 20) Sennrich R, Haddow B, Birch A. Neural machine translation of rare words with subword units. arXiv:1508.07909 [Preprint] 2016. Available from: https://arxiv.org/abs/1508.07909

[Comments from Reviewer 1]

(No Response)

Author response: The reviewer 1 did not give any comment.

[Comments from Reviewer 3]

The author (s) improved the paper based on the received comments. the paper can be published after improving language.

Author response: In response to the above comments, all spelling and grammatical errors pointed out by the reviewers have been corrected. We highlighted all the changes within the revised manuscript.

---

## [Editor Report · Decision Letter 2]

5 Jan 2023

Building an annotated corpus for automatic metadata extraction from multilingual journal article references

PONE-D-22-06759R2

Dear Dr. Kong,

We’re pleased to inform you that your manuscript has been judged scientifically suitable for publication and will be formally accepted for publication once it meets all outstanding technical requirements.

Kind regards,

Sanaa Kaddoura

Academic Editor

PLOS ONE

Additional Editor Comments (optional):

Reviewers' comments:

<quillbot-extension-portal></quillbot-extension-portal>

---

## [Editor Report · Acceptance letter]

11 Jan 2023

PONE-D-22-06759R2 

Building an annotated corpus for automatic metadata extraction from multilingual journal article references 

Dear Dr. Kong:

I'm pleased to inform you that your manuscript has been deemed suitable for publication in PLOS ONE. Congratulations! Your manuscript is now with our production department. 

Kind regards, 

on behalf of

Dr. Sanaa Kaddoura 

Academic Editor

PLOS ONE